# Phytochemical Composition and Antioxidant Activity of a *Viscum album* Mother Tincture

**DOI:** 10.3390/plants14172762

**Published:** 2025-09-04

**Authors:** Paola Imbimbo, Carolina Fontanarosa, Angela Amoresano, Daria Maria Monti, Gennaro Battaglia, Marcello Nicoletti, Michele Spinelli, Gerhard Schaller, Vincenzo Rocco

**Affiliations:** 1Department of Chemical Sciences, Complesso Universitario di Monte Sant’Angelo, University of Naples Federico II, Via Cinthia 21, 80126 Naples, Italy; paola.imbimbo@unina.it (P.I.); carolina.fontanarosa@unina.it (C.F.); angela.amoresano@unina.it (A.A.); mdmonti@unina.it (D.M.M.); gennaro.battaglia@unina.it (G.B.); michele.spinelli@unina.it (M.S.); 2Consorzio Interuniversitario Istituto Nazionale Biostrutture e Biosistemi, Viale Medaglie D’Oro, 00136 Rome, Italy; 3Department of Environmental Biology, Foundation in Unam Sapientiam, Sapienza University of Rome, 00185 Rome, Italy; 4Iscador AG Switzerland, Kirschweg 9, 4144 Arlesheim, Switzerland; gerhard.schaller@iscador.ch; 5CeMON s.r.l., Viale Gramsci 18, 80122 Naples, Italy; v.rocco@cemon.eu

**Keywords:** *Viscum album*, mistletoe, phytochemistry, antioxidant, mother tincture extract, toxicity

## Abstract

In the last decades, extracts of *Viscum album* L., commonly known as European mistletoe, have attracted increasing interest for their immunomodulatory, anti-inflammatory and antioxidant activities. Nowadays, they are mainly used in complementary cancer treatments. A targeted LC-MRM-MS was selected to determine the chemical composition and the activities of a *V. album* homeopathic mother tincture (MT#39998). Results showed a complex chemical composition, which was compared with that of other similar extracts. The LC-MRM-MS data were confirmed and complemented by HPLC analysis. Viscotoxins content was evaluated because of their cytotoxicity. MT#39998 was tested for its cytotoxic and antioxidant effect, before and after viscotoxins removal. The composition of MT#39998 in viscotoxins was similar to that of other products already present in the market and its safety was confirmed by estimation of LD_50_ based on in vitro IC_50_ values (LD_50_ was >2 g/kg). The aim of this study is to report a case study on a plant extract. The study was based on the chemical composition, including the metabolome, and on the pharmacological data, including toxicity and antioxidant activities, to validate the current utilization.

## 1. Introduction

The pharmacological and medical use of *Viscum album* L., commonly known as European mistletoe, is controversial [1,2,3,4]. Anciently, it was widely utilized and considered a panacea. Later and for a long time, the only acknowledged property of the plant seemed to be its toxicity [5,6,7]. About a century ago, thanks to anthroposophic medicine, the extract was used in oncological treatments [7,8,9]. Nowadays, *V. album* (*Viscum album)* extracts are used in central Europe as a co-adjuvant in the medical treatment of cancer, reflecting their immunostimulant, anti-inflammatory and antioxidant properties [10,11,12,13,14,15,16,17]. In several clinical trials, the value of the extracts antioxidant and anti-inflammatory activities in supporting the action of primary anti-tumoral drugs has been reported [18,19,20], with beneficial effects on the quality of life of hospital patients [21,22,23,24,25,26,27].

Given the general consensus on the reported beneficial activities, supported by consistent literature data [28,29,30,31,32,33], many extracts, with different phytochemical composition, are already on the market [34,35,36,37,38,39,40,41,42,43]. These extracts show several differences in their composition, probably due to different host trees and to various preparation methods, which may affect both the quality and the quantity of the constituents of the extracts. Several phytochemical methods have been used to determine the chemical composition of the extracts, the results of which are not completely comparable [44]. We report a comprehensive characterization of a *Viscum album* homeopathic mother tincture (MT#39998). First, the chemical composition of the preparation of *Viscum* extracts was analyzed using a metabolomic approach. The Targeted LC-MRM-MS methodology was selected, as it is able to reveal the presence of a wide range of compounds, from viscotoxins and viscolectins, typical of the composition of mistletoe extracts [38,39,40,43,44,45,46], to several secondary metabolites, including flavonoids and terpenes [41,47,48,49,50,51]. This analysis showed a very complex proteome, with several interesting and promising peptides, in accordance with the recent literature [52,53,54,55,56,57,58,59,60,61,62], as well as several antioxidant constituents. An MTT assay was performed on several cell lines and a well-established cell model was used to define the antioxidant activity. Intracellular ROS levels and GSH were evaluated in the presence of UVA-induced oxidative stress. As viscotoxins have a controversial effect on human health, different *Viscum* products present on the European market were evaluated for their viscotoxin levels. In order to better define the role of viscotoxins, the MT was deprived of viscotoxins and tested for its toxicity effect and antioxidant activity.

## 2. Materials and Methods

### 2.1. Viscum Extracts and Products

Table 1 reports the *Viscum* plant materials and marketed products utilized in this study, with some basic information. Samples were selected in accordance with traceability information concerning their production and origin. Viscum MT extracts (MT#39998, MT#40286 and MT#40293, provided by Herbamed AG, Bühler, CH) are standardized homeopathic mother tinctures (MTs) obtained by maceration from *Viscum album* L. (Santalaceace) fresh leafy shoots and ripe fruits of the *V. album* plant, which was grown on apple trees (*Malus domestica* Borkh.). The apple tree is a common host of mistletoe plants. After collection, the raw plant material was immediately cleaned with water and extracted following the Homeopathic German Pharmacopoeia (GHP), using method 2a [63] which has been incorporated into the European Pharmacopoeia method 1.1.3 (HAB 2a: MOTHER TINCTURES AND LIQUID DILUTIONS). Method 1.1.3/2a is used for fresh herbal drugs containing generally less than 70% expressed juice and more than 60% moisture (determined by loss on drying (2.2.32)) and no essential oil or resin. Briefly, once comminuted, a sample of the raw material (2.00–5.00 g) is taken and the loss on drying determined at 105 °C for 2 hrs. To the comminuted herbal drug immediately not less than half the mass of ethanol (90% *v*/*v*) is added and allowed to stand in well-closed containers. To calculate the quantity of ethanol required for maceration, the following formula is used: $\frac{m\times T}{100}$ where ***m*** is the mass of raw material in kg and ***T*** is the percent loss on drying. This expression is used to calculate the final amount (***A2***), in kilograms, of ethanol (90% *v*/*v*) required for the mass (***m***) of raw material. The value is than subtracted to the already added ethanol and the difference added to the mixture. The mixture is macerated for at least 10 days, filtrated and the dry residue is calculated. The formula: $\frac{m\times \left(N_{x}-N_{0}\right)}{N_{0}}$, where ***m*** is the mass of the filtrate, ***N*_0_** is the dry residue indicated for the standardized mother tincture, MT (>7.5%), and ***N_X_*** is the actual dry residue used to adjust the final volume of MT with 50% ethanol. Each preparation is controlled for density (0.955–0.975), dry residue (>7.5%), organolectic properties (the MT is a brown-yellow to brown-red liquid with a slightly aromatic odor) and the presence of specific signals on a TLC to compare with specific standards (routine, hyperoside and caffeic acid).

The Homeopathic Pharmacopoeia establishes that fresh leafy shoots must be collected with their ripe fruits but does not indicate a specific harvesting time. However, since the literature shows [42,43,50] that some *Viscum* extracts may differ in the harvesting period, the collection occurred in November for MT#39998, February for MT#40286 and in March for MT#40293. The *Viscum*-MT leaves sample was obtained only from *V. album* leaves of the same tree, collected in September. Iscador M#442 is a water-based European mistletoe medicine commercially available in Switzerland, Vischio Centofiori (VC#011473), Coragil and Rubaxx arthro are homeopathic medicines based on *Viscum* mother tincture, commercially available in Germany, and Vischio Centofiori and Vischio Biokyma are Italian food supplements, with the first supplied as a glycerol-alcoholic solution and the second as a hydro-alcoholic one. The MT samples were selected to compare the composition and activity of products collected during different periods or corresponding to different parts of the *Viscum* plant from the same host or subjected to different preparation and production methods. The *Viscum* MT#39998 was used as the target sample of the validation study. It was also compared with several *Viscum* products available on the market, in a range of phytomedicines and food supplements. In this way, it was possible to check the influence on the composition of pre- and post-harvesting factors, as well as evidence of differences between the *Viscum* MT and products already present in the market and so far considered safe and useful. Furthermore, to obtain an MT depleted of viscotoxins (herein named vd-MT), the extract was treated in accordance with Stein et al. [64], whereas the sample MT#39998 was treated as described by Holandino et al. [65].

Briefly, the MT was allowed to flow under vacuum through a Bakerbond, carboxylic acid, wide-pore SPE column (J.T. Baker—Avantor Science Central, Phillipsburg, New Jersey, USA), in order to retain the viscotoxins and other low molecular weight proteins. The SPE column was previously washed with methanol and water and subsequently equilibrated with 5 mL of a 200 mM ammonium solution. Aliquots from 0.5 to 2 mL of MT were added to the column and the pH was adjusted to 7.0–7.5. Then, to elute the retained viscotoxins (pv purified viscotoxins), 3 mL of water was used to rinse the column and the samples were finally eluted with 5 mL of 0.4 M acetic acid. The vd-MT, as well the other MTs, were stored at room temperature.

### 2.2. Chemicals and Reagents

Urea, dithiothreitol (DTT), trypsin, iodoacetamide (IAM) and ammonium bicarbonate (AMBIC) were purchased from Sigma-Merck (Milan, Italy). Formic acid, methanol, acetonitrile, analytical grade hydrochloric acid, ethyl acetate and n-hexane were obtained from J.T. Baker (Phillipsburg, NJ, USA). The labeled peptides were purchased from Thermo Fisher Scientific, i.e., viscotoxin A, A2, A3(1), A3(2), A3(3), C1,1-PS. Pipette tips were homemade using Empore octadecyl C18 47 mm disks obtained from Supelco.

### 2.3. Cell Culture

Immortalized murine fibroblasts BALB/c-3T3, murine fibroblasts transformed with virus SV40 (SVT2) and human cervix carcinoma cells (HeLa) were obtained from ATCC, whereas immortalized human keratinocytes (HaCaT) were obtained from Innoprot (Biscay, Spain). All cells were cultured in Dulbecco’s Modified Eagle’s Medium (DMEM) (Sigma-Aldrich, St Louis, MO, USA), supplemented with 10% fetal bovine serum (HyClone), 2 mM l-glutamine and antibiotics, all from Sigma-Aldrich, and maintained under a 5% CO_2_ humidified atmosphere at 37 °C.

### 2.4. MTT Assay

The effects on cell viability of the mother tincture (MT#39998) and vd-MT were evaluated on different cell lines. HaCaT and HeLa cells were seeded in 96-well plates at a density of 2 × 10^3^/well, BALB/c-3T3 at a density of 3 × 10^3^/well, whereas SVT2 cells were seeded at a density of 1.5 × 10^3^/well. After 24 h, cells were incubated with increasing concentrations of MT#39998 and vd-MT (from 0.05 to 5% *v*/*v*). Upon 48 h incubation, cell viability was assessed by MTT as previously reported [66]. Cell survival is expressed as the percentage of viable cells in the presence of samples compared with control cells (represented by the average obtained between untreated cells and cells supplemented with the highest concentration of buffer).

### 2.5. Antioxidant Activity

HaCaT cells were seeded at a cell density of 2 × 10^4^ cells/cm^2^. After 24 h, cells were incubated with 0.5% *v*/*v* of MT or vd-MT for 2 h. Then, cells were exposed to UVA radiation (100 J/cm^2^) as a source of oxidative stress. Immediately after stress induction, ROS production was estimated by DCFDA assay, as reported by Liberti [67] and intracellular GSH levels were determined by DTNB assay, according to the protocol reported by Petruk et al. [68].

### 2.6. HPLC Profiling

Samples were diluted in 0.2 M acetic acid, centrifuged at 10,000 rpm for 10 min, and the supernatant was loaded and analyzed on an Agilent 1200 HPLC system (Santa Clara, CA, USA) equipped with an Agilent Zorbax Eclipse Plus C18 4.6 × 100 mm, 3.5 µm and a UV detector (210 nm). The mobile phase consisted of eluent A (0.1% TFA in water) and eluent B (0.1% TFA in acetonitrile/water 60/40). The gradient ranged from 0 to 7 min at 10% B, from 7 to 7.5 min at 30% buffer B for 1 min, from 8.8 to 9.5 min at 50% eluent B for 1 min, and from 10.5 to 11.5 min at 70% eluent B for 3 min, followed by 70% to 90% eluent B in 5 min at a flow rate of 1 mL per minute.

### 2.7. Preparation of Samples by Solution Digestion

Samples were solubilized in 50 µL 6 M urea and 50 mM AMBIC, treated with 400 µL cold acetone and kept at −20 °C for 30 min. This process was repeated three times. Subsequently, samples were centrifuged at 4 °C for 10 min at 7000 rpm to promote protein precipitation. Proteins were quantified by Bradford assay [69]. Proteins were reduced with 20 mM DTT, and incubated for 10 min at 95 °C. After cooling the protein solution to room temperature, the cysteines were alkylated by the addition of IAM 40 mM, followed by incubation in the dark for 30 min. The remaining IAM was quenched with 20 mM DTT for 60 min in the dark. Before adding the enzyme, urea was diluted with AMBIC 50 mM to a concentration of 1 M. Trypsin was added in a 1:50 enzyme/protein ratio and incubated for 16 h at 37 °C. Digestion was stopped by adding 1% formic acid. Custom-made chromatographic microcolumns were used for desalification of samples prior to mass spectrometric analysis. Sample loading, washing, and elution were performed by centrifugation at 5000× *g* for 2 min. Each tip was used only once to avoid contamination. A quantity of 100 µL of 80% ACN, 80% ACN/20% HCOOH (5%) and 0.5% HCOOH was used to equilibrate a custom-made chromatographic microcolumn. Samples dried were dissolved in 50 µL of 0.5% HCOOH and loaded. Before loading, 100 µL of 0,5% HCOOH was used for washing. A quantity of 50 µL of both ACN/HCOOH (0.5%) 1:1 and ACN/HCOOH (0.5%) 8:2 was used for elution of peptides. Samples were dried in speed-vac and resuspended in 2% ACN and analyzed by both untargeted and targeted tandem mass spectrometry LC-MS/MS analysis using a Xevo TQ-S (Waters) equipped with an ionKey UPLC Microflow Source coupled to a UPLC Acquity System (Waters).

### 2.8. Untargeted LC-MS/MS

After digestion, peptide samples were loaded via an autosampler (Surveyor MS Pump Plus and Micro AS) onto a Michrom C18 Captrap and were then introduced directly into a Orbitrap LTQ Velos MS (Thermo Fisher Scientific, Surrey, UK) via a fused silica C18 capillary column (Nikkyo Technos CO, Tokyo, Japan) and a nanoelectrospray ion source. The mobile phase comprised 0.1% aqueous formic acid (buffer A) and 100% acetonitrile with 0.1% formic acid (buffer B). The gradient ranged from 5% to 30% buffer B for 95 min, followed by 30% to 60% B for 15 min and a step gradient to 85% B for 5 min with a flow of 0.42 μL min^−1^. Finally, the system was returned to the initial conditions of 5% B. FTMS full-scan mass spectra (from 450 to 1600 *m*/*z*) were acquired with a resolution of r = 60,000. This was followed by data-dependent MS/MS fragmentation in centroid mode of the most intense ion from the survey scan using collision-induced dissociation (CID) in the linear ion trap: normalized collision energy 35%; activation Q 0.25; electrospray voltage 1.5 kV; capillary temperature 200 °C; and isolation width 2.00. The MS/MS scan event was repeated for the top 20 peaks in the MS survey scan; the targeted ions were then dynamically excluded for 30 s. Singly charged ions were excluded from the MS/MS analysis; Xcalibur software version 2.1.0 SP1 build 1160 (Thermo Fisher Scientific, UK) was used for data acquisition.

### 2.9. Mascot Identification

The acquired MS/MS spectra were transformed in *m*/*z* Data (.XML) format and used for protein identification with a licensed version of MASCOT software (www.matrixscience.com, access on 1 March 2023) version 2.4. with 10 ppm MS tolerance and 0.6 Da MS/MS tolerance; peptide charge from +2 to +3. Carboxyamidomethylation of Cys as a fixed modification was inserted, but possible oxidation of methionines and formation of pyroglutammic acid from glutamine residues at the N-terminal position of peptides were considered as variable modifications to query the SwissProt databases.

### 2.10. Targeted LC-MRM-MS

A mix of standard heavy peptides (synthesized by Invitrogen-Life technologies; see Appendix A for the sequence) was prepared at 100 nM in DMSO, divided into aliquots, and stored at −80 °C. After digestion, the mix was spiked into the individual samples at 250 fmol/μg of extract protein for analytes targeted by LC−MRM, and the samples were desalted as described in the Preparation of Standard Solutions and Samples section (below). Spike levels were high enough above the LOQ so as not to contribute unnecessarily to the assay CV. Quantitative LC−MRM-MS data were collected using the method described below. LC-MS/MS analyses were performed using a Xevo TQ-XS tandem-quadrupole mass spectrometer (Waters) equipped with an ionKey UPLC Microflow Source coupled to a UPLC Acquity System (Waters). All peptide mixtures were analyzed using the same chromatographic conditions. For each run, 1 μL peptide mixture was injected and separated on a TS3 1.0 mm × 150 mm analytical RP column (Waters, Milford, MA, USA) at 45 °C, with a flow rate of 3 μL/min using 0.1% aqueous HCOOH (LC-MS grade) as eluent A and 0.1% HCOOH in ACN as eluent B. Peptides were eluted (starting 1 min after injection) with a linear gradient of eluent B in A from 7% to 95% in 55 min. The column was re-equilibrated at initial conditions for 4 min. The MRM mass spectrometric analyses were performed in positive ion mode using an MRM detection window of 0.5−1.6 min per peptide; the duty cycle was set to automatic and dwell times were minimal at 5 ms. The cone voltage was set to 35 V. All the transitions related to the heavy and light viscotoxins peptides selected for the analysis are reported in Appendix A.

### 2.11. Targeted LC-MS/MS Analysis

#### Preparation of Standard Solutions and Samples

Standard stock solutions (Sigma Aldrich, gallic acid 91215, vitamins mixture 748284—1 mL, amino acids standard mixture 71, 21, 85—2 mL, phytol crm40375) were prepared by adding 1.00 mL aliquots of each analyte to a 10 mL volumetric flask and bringing the standard to volume with methanol to yield a standard solution with 1000 µg/L of each analyte. The stock solutions were stored at −20 °C until analysis. Quantitative analysis was performed by construction of calibration curves for a set of standard molecules selected for the different classes of analytes under investigation. Samples were diluted 1:10 in the methanol solvent and filtered and centrifuged at 10,000× *g* for 10 min. The supernatant was then directly transferred into the HPLC auto sampler. The supernatant (1 μL) was analyzed using an AB-sciex 5500 QTRAP^®^ system with an HPLC chromatography system Exion LC™. The mobile phase for polyphenols analysis was generated by mixing eluent A (0.1% formic acid in water) and eluent B (0.1% formic acid in acetonitrile) and the flow rate was 0.200 mL/min. The chromatographic gradient was 20% to 90% in 4 min, held for 2 min, then returned to 20% in 1 min. Vitamin and amino acid analysis was carried out using 0.1% formic acid in water with 20 mM ammonium formate (mobile phase A) and 0.1% formic acid in methanol with 20 mM ammonium formate (mobile phase B) and the flow rate was 0.200 mL/min. The chromatographic gradient was as follows: 10% B for 1 min, to 50% B in 1 min, held for 2 min, then to 90% in 1 min, held for 2 min, then returned to 10% in 2 min. Tandem mass spectrometry was performed using a Turbo VTM ion source operated in positive and negative ion mode, and the multiple reaction monitoring (MRM) mode was used for the selected analytes. Appendix A provide a list of the precursor ions, product ions, collision energy and declustering potential parameters. The extracted mass chromatogram peaks of the metabolites were integrated using Skyline software for data processing. For each molecule, specific transition precursor ions/fragment ions were selected (Appendix A). Quantitative analyses were performed using the external standard method by calibration and interpolation.

### 2.12. Untargeted GC-MS Analysis

#### Fatty Acids Analysis

A quantity of 1 mL of CHCl_3_, 850 µL of MeOH and 150 µL of H_2_SO_4_ were added to the dried samples. The methylation reaction was conducted for 16 h at 90 °C. The reaction was stopped using 2 mL of sodium carbonate, and then the lower organic phase was collected and analyzed by GC-MS. GC-MS analysis was performed by a 7820A (Agilent Technologies, Santa Clara, CA, USA) with a DB-5 capillary column (30 m × 0.25 mm × 0.25 µm film thickness) (Agilent Technologies). The injector, ion source, quadrupole and the GC–MS interface temperature were 230, 230, 150 and 280 °C, respectively. The flow rate of the helium carrier gas was kept at 1 mL/min. A quantity of 1 µL of derivatized sample was injected with 3 min solvent delay time and a split ratio of 10:1. The initial column temperature was 40 °C held for 2 min, ramped to 150 °C at a rate of 15 °C/min and held for 1 min, then finally increased to 280 °C at a rate of 30 °C/min and kept at this temperature for 5 min. Ionization was carried out in electron impact (EI) mode at 70 eV. The MS data were acquired in full scan mode from *m*/*z* 40–400 with an acquisition frequency of 12.8 scans per second. The identification of compounds was confirmed by injection of pure standards (FAME mixture for fatty acids quantification, Sigma Aldrich) and comparison of the retention time and corresponding EI MS spectra. The contents of fatty acids were calculated using the external standard method by calibration and interpolation.

### 2.13. Estimation of LD_50_ Based on In Vitro IC_50_ Values

The in vivo LD_50_ of acute oral toxicity was estimated from in vitro IC_50_ using the following formula:$$Log\;LD_{50}=0.372\;\times log\;IC_{50}\;\left(µg/\mathrm{mL}\right)+2.024$$ as reported in [70] and adapted for molecules with unknown molecular weight. In an independent study, the authors verified that for low or non-toxic substances, the formula is reliable for estimation of oral LD_50_ [71].

### 2.14. Statistical Analyses

All the experiments were performed in triplicate. The results are presented as the mean of results obtained after three independent experiments (mean ± S.D.) and compared using one-way ANOVA according to Bonferroni’s method (post hoc) using GraphPad Prism for Windows, version 6.01 (Dotmatics, San Diego, CA, USA).

## 3. Results

### 3.1. Viscum album Mother Tincture Chemical Characterization

Two batches of *Viscum album* MTs were used to obtain reliable chemical composition analyses. Untargeted and targeted LC-MS/MS, along with untargeted GC-MS analysis, were utilized to characterize the *Viscum album* MTs and quantify their components. As reported in Table 2 and Figure 1, the extracts were very rich in phenolic compounds, amino acids, phytols, fatty acid esters and sugars. Phenolic compounds predominated and therefore were investigated further. Interestingly, the MTs were very rich in antioxidants, representing almost 50% of the molecules present in the extract (phenolic compounds and phytols). Among the amino acids, arginine was the most abundant (>60%, data reported in Appendix A). Esters of alpha-linoleic acid were also abundant. These results are in accordance with previous reports [72]. Interestingly, the two analyzed MTs showed a difference in the abundance of their constituents, probably due to a different harvesting time. MT#39998 was collected in November when the fruits are ripe, whereas in February (MT#40293), the plant starts to flower.

#### *Viscum album* Phenolic Compounds

Table 3 shows the detected phenolic compounds and their concentrations in the extracts. Interestingly, even though many of these compounds have already been found in *Viscum album malus* [73], others are new, such as diosmetin, vitexin and epigallocatechin-3,3′-digallate. Quercetin, naringenin, naringin and myricetin represent a group of molecules, which, besides viscotoxins, can contribute to the antioxidant and anticarcinogenic properties of the plant [74,75,76,77]. The data reported in Table 3 confirm differences in the amount of the phytoconstituents due to the collection time, as detected phenolic compounds were more abundant in MT#40293 than MT#39998, with some exceptions. For example, naringenin, coumaric acid and *cis*-resveratrol-3-*O*-glucoside were more abundant in the November-harvested MT than in February. The presence of these compounds, together with chlorogenic acid and caffeoylquinic acid, can be easily related to antioxidant activity. There is a significant presence of flavonoids, including as glycosides, with different types of conjugation and functional substituents observed.

### 3.2. Protein Analysis of Viscum album Extracts

*Viscum album* MTs, as well as other selected different products commercialized in Europe, were analysed by HPLC and by HPLC-MS/MS in MRM to compare the viscotoxin contents.

#### 3.2.1. HPLC Analysis

MT#39998 viscotoxins content was quantified in the Iscador laboratory by HPLC (Table 4). The HPLC profiles of MT#39998, of the purified viscotoxins and of a known product (Iscador, I-M) were subsequently compared (see Figure 1A–C). The samples showed very different profiles, evidencing a complex mix of proteins (see below and Appendix A) and vitamins (Appendix A). A mother tincture deprived of viscotoxins (vd-MT) was prepared following the Holladino et al. protocol [65]. As expected, the viscotoxins were successfully removed, but other peptides were still present. Therefore, to obtain more precise and comparable data from the natural extracts/products we wanted to look at, we than utilized MS. In Figure 1B, it is possible to observe the absence of viscotoxin retention peaks on the right side of the graph.

#### 3.2.2. Untargeted Proteomics

The same amount of protein content (100 μg) tested in HPLC was used to carry out enzymatic digestion for the MTs untargeted proteomics experiments. Thus, proteins were hydrolysed, purified and then analyzed by HPLC-MS/MS in a high-resolution experiment. Tandem mass spectra were then submitted to database search. MASCOT identification revealed the presence of several proteins, reported in Table 5 and the Appendix A. Among them, viscotoxins and lectins were present. Interestingly, some peroxidase and superoxide dismutase peptides analogous to known plant sequences were found.

### 3.3. Multiple Reaction Monitoring Analysis of Viscotoxins

The above-reported results clearly demonstrate that viscotoxins are responsible for the well-known cytotoxicity of the *Viscum* extract, but do not impair antioxidant activity. Thus, the presence of viscotoxins was evaluated on different mother tinctures commercially available in the EU: Iscador, Rubax Artro, Coragil, Centofiori and Biokyma. Viscotoxins identification and quantification were obtained using a proteomics targeted approach, applying HPLC-MS/MS in MRM ion mode. Several unique peptides belonging to the target analyte were selected through in silico analysis using the Skyline software that provided the predicted best transitions and collision energy to generate maximal fragment intensities. The peptides defined by Skyline analysis were selected to develop an MRM method for each sample. Raw data obtained from LC-MRM MS/MS were analyzed with Skyline targeted mass software and the integrations obtained were manually inspected to ensure correct peak detection, absence of interference and accurate integration. As an example, Figure 2 reports the total ion current in the MRM of Viscotoxin A2, showing transitions of peptide LTGAPCPTCAK. For quantitative analysis, a calibration curve for each target peptide was obtained. Specificity was confirmed by equivalent retention time and the relative areas of the light and heavy transitions. The proteins of interest were quantified by interpolation of the relative areas of the ionic currents of the selected transitions on the constructed calibration curve using the external standard method.

As shown in Table 6, many viscotoxin isoforms were found with similar content, all of which were present in the analyzed samples, except for Viscotoxin C, which was not present in samples Co #82102, RA #210071, VB #217. In particular, Viscotoxin A3 was the most represented, followed by A2, A1, B and A1PS, with just one exception in sample MT#40286 where A1 > A2. Food supplements also contain viscotoxins, though in lower amounts.

### 3.4. Effect of MT#39998 and vd-MT on Cell Viability

The effect of MT#39998 and vd-MT on cell viability was evaluated on two immortalized cell lines, HaCaT and BALB/c-3T3, and two cancer cell lines, HeLa and SVT2. At 24 h after plating, cells were incubated with an increasing amount of both extracts (from 0.05 to 5% *v*/*v*). Upon 48 h incubation, cell viability was assessed using the MTT assay. The results are reported in Figure 3, including IC_50_ values (i.e., the concentration of extracts able to reduce cell viability to 50%) in Table 7. The results indicate that the MT#39998 (Figure 3A,B) has a cytotoxic effect on both immortalized and cancer cell lines. In particular, the IC_50_ values were 2.6% and 1.6% on HaCaT and BALB/c-3T3 cells, respectively, whereas they were 1.3% and 2.1% on SVT2 and HeLa cells, respectively. The vd-MT (Figure 3C,D) was found to be toxic only on SVT2 at the highest concentration tested. Overall, these results indicate that viscotoxins contribute most to the cytotoxicity of the preparation.

### 3.5. Estimation of LD_50_

Conventionally, the safety of substances is determined using in vivo acute oral toxicity tests, by determining the dose that is lethal in 50% of the tested animals (LD_50_), generally rats [78]. According to the Interagency Coordinating Committee on the Validation of Alternative Methods [79], an LD_50_ value for rats can be estimated from the IC_50_ value obtained on a cell-based cytotoxicity assay. This approach enables the estimation of a starting dose closer to the actual LD_50_ value for in vivo acute oral toxicity studies, thus reducing the total numbers of animals to be used. Even though widely employed in traditional and homeopathic medicines, data on the oral LD_50_ of *V. album* MTs or related oral products are lacking. The expected LD_50_ values for the analyzed MTs were determined using the formula reported in the Materials and Methods section and the results are shown in Table 8. For all the four cell lines, the LD_50_ was >2 g/kg, which suggests the absence of toxic effects for the MT#39998. The viability data obtained from vd-MT indicate that viscotoxins are responsible for the residual cytotoxicity of the extract. On the other side, the in vitro data were obtained using doses that are very high compared to standard doses used in homeopathy or suggested in supplements (10–40 drops per daily dose corresponding to 0.5–2 mL/dose for an average weight person of 70 kg, or 1.92 g/70 kg or 27.43 mg/kg_b.w._). Oral LD_50_ values have been measured for *Viscum* methanolic extracts, showing an LD_50_ > 2 g/kg, indicating the total biocompatibility of these extracts. Similar results have been found for aqueous extracts [80].

The overall results suggest that products on the market which are based on *Viscum* extracts are safe.

### 3.6. Antioxidant Activity

Finally, the protective effect of MT#39998 and vd-MT against oxidative stress was evaluated on a cell-based system. Immortalized human keratinocytes (HaCaT cells) were chosen, as they represent the external layer of the skin. As a stress source, a UVA lamp was used. The UVA lamp system employed is normally used in the nail industry. Cells were treated with 0.5% of MT#39998 or vd-MT for 2 h and then oxidative stress was induced by UVA irradiation (100 J/cm^2^). Immediately after UVA irradiation, intracellular ROS levels were determined by H_2_-DCFDA as the probe. As shown in Figure 4A, in the absence of any treatment, when cells were irradiated by UVA, a significant (*p* < 0.05) increase in ROS levels was observed (200%), whereas the treatment with either MT#39998 or vd-MT in the absence of oxidative stress did not alter the physiological production of ROS. Interestingly, when cells were pretreated with either MT#39998 or vd-MT prior to UVA exposure, inhibition of ROS production was observed. To confirm the protective effect, intracellular GSH levels were measured using DTNB assay. The results are shown in Figure 4B. In the absence of treatment, a significant (*p* < 0.001) decrease in GSH levels was observed, and a slight oxidation effect was also detected when cells were treated with vd-MT. It is of note that pretreatment with both extracts was able to prevent the oxidation of GSH, thus confirming that MT#39998 and vd-MT are both able to protect cells from oxidative stress and that viscotoxins are not responsible for the antioxidant activity.

## 4. Discussion

Besides their use in different traditional medicines and oncology, *Viscum album* extracts are present in many healing products for several clinical applications, i.e., to lower hypertension, reduce triglycerides and other cardiovascular conditions, for arthritis, secondary menopause symptoms, as well to treat all the pathologies related to oxidative stress and inflammation [81,82,83,84,85,86]. These extracts, in contrast to the above-mentioned cancer-related products which are injected, are available on the market as liquids for oral use. Therefore, the evaluation of *Viscum*-based market products, including homeopathic products and plant extracts, is extremely important and necessary. In general, the phytochemical analyses of MTs evidenced a similarity in the composition of the selected products, with a prevalence of phenolic compounds and variation in their concentration depending on the seasonal harvesting time. The MRM findings are consistent with the results obtained by Melo et al. using an untargeted approach [87,88].

The antioxidant property of MTs was evidenced, as well as the constituents responsible for the biological activity identified. The IC_50_ data were obtained on four cell lines, with values increasing from 1.3 to 2.6, depending on the cell line. The general oral safety of the products was indicated by calculation of the theoretical LD_50_ from IC_50_ data, showing an LD_50_ > 2 g/kg. The MT#39998 showed a remarkable ability to protect cells against UVA stressed cells, as indicated by the inhibition of ROS production, and the high levels of GSH. These effects are consistent with the homeopathic use of the MT for coronary artery disease and hypertension [89,90,91,92,93]. More recently, it has been suggested that *Viscum album* MT be used in cancer treatments for its ability to interrupt the Warburg effect [94].

Despite the utilization of *Viscum* products in several clinical treatments and the large positive literature about the absence of serious negative effects, as well as positive reports on the quality of life of patients, some concerns about *Viscum* toxicity and the constituents responsible remain. In this research, the metabolic content of the selected extract MT#39998 was extensively analyzed and related to several effects, including toxicity, in order to extend the extant literature, which has mainly focused on antitumor effects or the chemical composition. This is the first time that such a comprehensive approach has been applied to a homeopathic product.

The chemical composition of the MT#39998 was precisely determined by utilization of hyphenated methods, such as targeted LC-MRM-MS, including attention to different types of constituents, from sugars to flavonoids. The extracts of *Viscum* were observed to be very complex and not limited to low molecular weight proteins. As confirmation, the same phytochemical analyses, when performed on other extracts obtained with similar methods, including marketed products, gave similar results in terms of quality and quantity. Therefore, MT#39998, selected as the main target, can be considered to be similar in composition to the other analyzed extracts.

The above results were fully confirmed by the HPLC analyses conducted, which focused on the percentage of microproteins, in particular viscotoxins, which are considered principally responsible for toxic effects. The results of the HPLC analysis were further informed by comparison with those for products widely available in Europe and utilized extensively over several decades. The results evidenced similar percentages of viscotoxins in all the selected extracts, as confirmed by HPLC analysis of MT#39998, in accordance with previous data [42] and the data obtained by the LC MRM MS analysis (Appendix A). The quantities observed in all cases were low and below the limit considered safe.

Despite the positive findings in the literature on the safety of the main marketed products, including comparison of the anti-proliferative effects on cancer and normal cellular lines, which evidence selective activity against cancer cell lines in accordance with clinical trials, an independent toxicologic study was performed on MT#39998. The alternative approach used [70] to the standard LD_50_ in vivo assay for acute toxicity involved using cytotoxicity data and extrapolating these to LD_50_. As far back as 1956, efforts to correlate these two assays was realized [95], reflecting the need to reduce animal sacrifice to appraise acute toxicity. Such a method has been challenged by Arnhild Schrage et al. [72] with several molecules of known toxicity. The result of these experiments showed a different ability of this method to predict LD_50_ depending on the level of toxicity of the substance. In synthesis, it was reliable for low- or non-toxic substances. In contrast, the authors suggest that for very toxic ones, the method is poorly effective. Our results, which relate to a non-toxic substance, substantially agree with those conclusions, as well as with the results from extracts obtained by similar solvents, such as methanol or water [80,82], and from the evidence on marketed or widely used products containing *Viscum* extracts, therefore indicating the safety of the mother tincture.

In our view, herbal treatments of unknown cytotoxicity, or for which data, as in the case of *Viscum*, are reported inconclusively in the literature imply a need to undertake in vivo toxicity assays. Based on the literature and the results of this study, it is possible to conclude that the toxicity of MT#39998 is comparable to the toxicity of the most-marketed extracts, which are already considered safe and widely used clinically in several countries. However, this conclusion cannot be extended to other extracts with higher concentrations in viscotoxins or that are very different in their chemical constitution. This implies that the quality control of products containing natural products must consider the safe range of concentrations of the constituents. Concentration data must be obtained with advanced phytochemical methods to provide comprehensive evidence of product composition.

## 5. Conclusions

In conclusion, the target MT extract was found to contain a rich variety of antioxidant substances, whose activity was confirmed. Its composition in terms of secondary and primary constituents was found to be similar to that of other products already present in the market. We observed slight variations in the quantity of antioxidants depending on the harvesting season. Some groups have found differences in the amount of viscotoxin and lectins [41] following harvesting time, possibly reflecting plant–host interaction. A similar dynamic may apply to the antioxidant molecules found, since they represent a barrier against the defences of the host plant. However, at the clinical level, since the season dependent variations in the concentration of the different phenolic compounds is quite low, it is difficult to assume that these differences may have an impact on product quality and/or efficacy.

Furthermore, the proteome analysis undertaken characterized the proteome of MT#39998. The reported data reflect the fact that the results of this kind of analysis depend on the sensitivity of the analytical method used. As such, the proteome must be considered to be a complex mixture of peptides of different size and amino acids. The current tendency to consider only the amount of viscotoxins must be revised and cannot be assumed to be a valid basis for evaluating the toxicity of *Viscum* products. Comparison of the collateral and negative effects, as reported in the literature, with already utilized products can be useful, as well as evidence of positive health efficacy. The propensity to infer that there are health dangers and acute toxicity risks of marketed *Viscum* products should be considered in light of the complete range of information and the potential consequences of losing useful health promoters.

## Figures and Tables

**Figure 1 plants-14-02762-f001:**
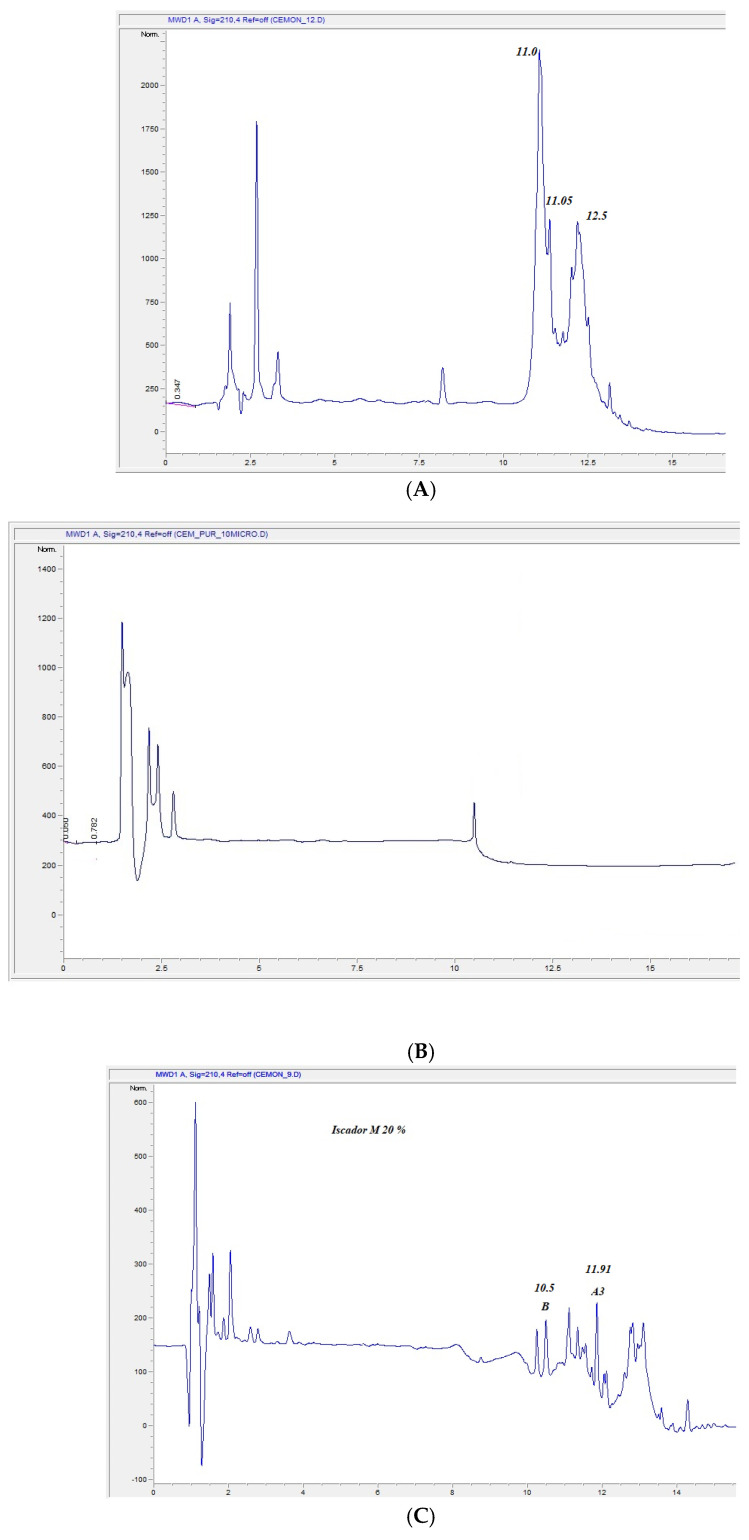
(**A**) HPLC protein profile of MT#39998. (**B**) HPLC profile of vd-MT#3998. (**C**). HPLC profile of Iscador M (I-M). The X-axes are the retention times, the Y-axes, the intensity.

**Figure 2 plants-14-02762-f002:**
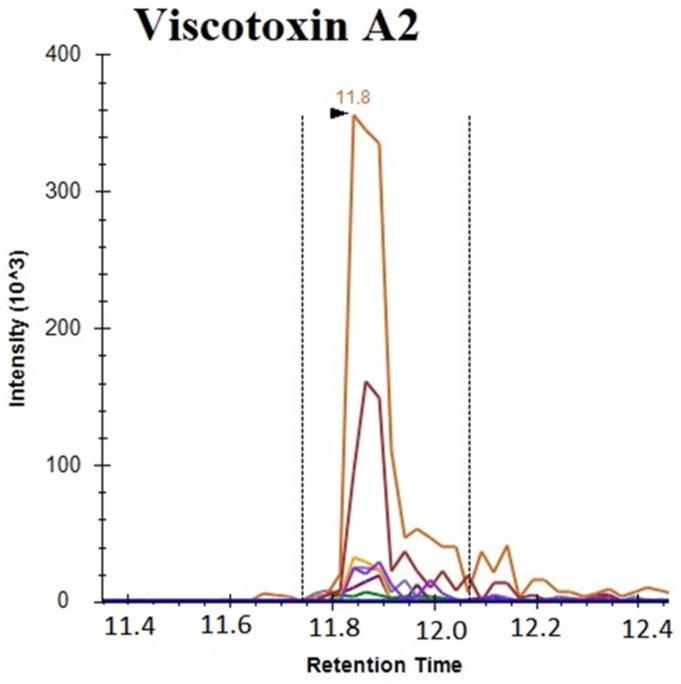
Total ion current in MRM of Viscotoxin A2 showing transitions of peptide LTGAPCPTCAK.

**Figure 3 plants-14-02762-f003:**
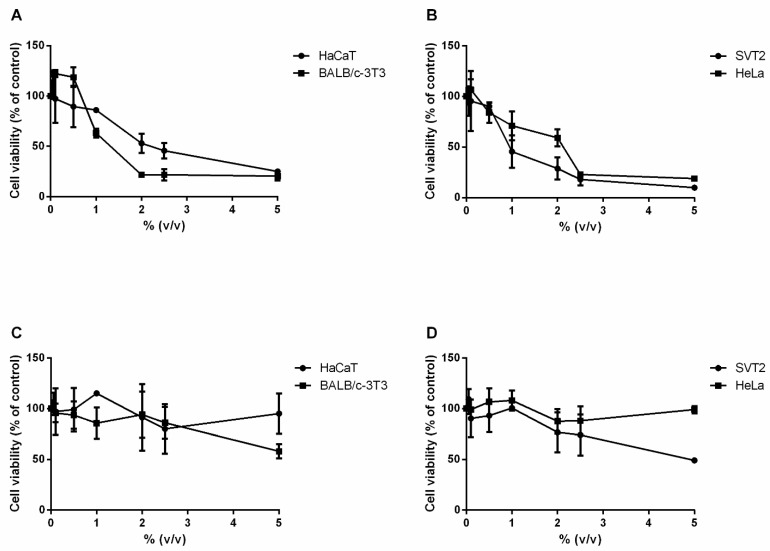
Effect on cell viability of MT-#39998 (**A**,**B**), and of vd-MT (from MT#39998) (**C**,**D**). HaCaT, BALB/c-3T3, SVT2 and HeLa cells were incubated with an increasing concentration of both extracts (0.05–5% *v*/*v*) for 48 h. Cell viability was assessed using the MTT assay. Values are expressed as means ± SD (n ≥ 3).

**Figure 4 plants-14-02762-f004:**
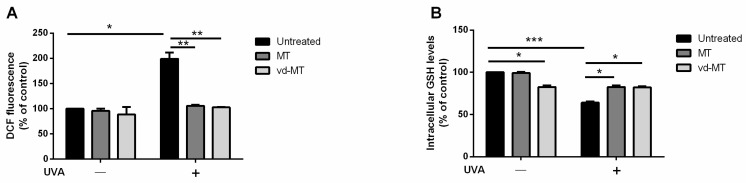
Protective effect against UVA-stressed HaCaT cells. Intracellular ROS levels were measured using DCFDA assay (**A**), and intracellular GSH levels were determined using DTNB assay (**B**). Cells were pretreated in the presence of 0.5% (*v*/*v*) MT or vd-MT for 2 h prior to UVA irradiation (100 J/cm^2^). ROS levels and GSH levels were measured immediately after UVA irradiation. Black bars refer to untreated cells, dark grey bars refer to cells incubated with MT#39998, and light grey bars refer to cells incubated with vd-MT, in the absence (−) or in the presence (+) of UVA stress. Values are expressed as percentages compared with untreated cells. * Indicates *p* < 0.05, ** indicates *p* < 0.01, *** indicates *p* < 0.001.

**Table 1 plants-14-02762-t001:** List of the reported *Viscum* products, including basic information.

Product	Name/Batch	Host Plant	Harvesting Time	Marketed as
*Viscum* MT#39998	#39998	*Malus domestica* Borkh.	November	Homeopathic starting material
*Viscum* vd-MT	#39998	*M. domestica*	November	N.A.
*Viscum* MT#40286	#40286	*M. domestica*	February	Homeopathic starting material
*Viscum* MT#40293	#40293	*M. domestica*	March	Homeopathic starting material
*Viscum*–MT leaves	#00293421	*M. domestica*	September	Starting material
Iscador M#442	M#442	*M. domestica*	Unknown	Anthroposophic medicine (Swiss)
Vischio Centofiori	VC#011473	Unknown	Unknown	Food supplement (Italy)
Coragil	Co#82102	Unknown	Unknown	Homeopathic medicine (Germany)
Rubaxx Arthro	RA#210071	Unknown	Unknown	Homeopathic medicine (Germany)
Vischio Biokyma	VB#217	Unknown	Unknown	Food supplement (Italy)

**Table 2 plants-14-02762-t002:** Total composition of MTs. Data are expressed in mg/L.

Component	MT#39998	MT#40293	AV
Phenolic compounds	66.87	96.63	81.75 ± 14.88
Phytols	20	30	25 ± 5
Sugars	14	15	14.5 ± 0.5
Fatty acid esters	16.8	21.09	18.95 ± 2.5
Fatty acids	1	1.5	1.25 ± 0.25
Amino acids	63.011	87.253	75.132 ± 12.12
Vitamins	0.56	0.68	0.62 ± 0.06

**Table 3 plants-14-02762-t003:** Phenolic compound analysis. Data are reported in mg/L.

Phenolic Compound	MT#39998	MT#40293
Vitexin	0.07	0.07
5-Caffeoylquinic acid	0.17	0.2
Neochlorogenic acid	20.04	34.11
Catechin	0.03	0.05
Chlorogenic acid	32.88	48.95
*cis*-Resveratrol-3-*O*-glucoside	0.15	0.12
Coumaric Acid	0.92	0.65
Diosmetin	2.68	3.08
Epigallocatechin-3,3′-digallate	0.01	0.01
Epigallocatechin-3-gallate	0.01	0.01
Eriodictyol	0.08	0.14
Ethyl gallate	0.03	0.04
Ferulic acid	0.79	1.15
Gallic acid	0.03	0.03
Genistein	0.01	0.01
Genistin	0.07	0.04
Hesperetin	0.04	0.07
Isorhamentin-3-O-neohesperidoside	0.69	0.69
Isorhamnetin	0.05	0.06
Kaempferol	0.16	0.18
Linarin	0.03	0.04
Luteolin	0.34	0.02
Luteolin-7-O-glucoside	0.13	0.17
Myricetin	0.01	0.02
Myricetin-3-O-glucoside	0.18	0.19
Naringenin	4.16	2.46
Naringenin-7-O-neohesperidoside	0.02	0.03
Naringin	0.78	1.38
Nicotinflorin	0.04	0.05
Orientin	0.17	0.19
Piceatannol	0.01	0.01
Procynadin C1	0.02	0.02
Protocatechuic acid	1.35	1.48
Quercetin	0.1	0.14
Quercetin-3-O-rutinoside	0.02	0.02
Rutin	0.53	0.64
Theaflavin-3,3-digallate	0.01	0.01
Theaflavin-3-gallate	0.01	0.01
Vanillic acid	0.05	0.09

**Table 4 plants-14-02762-t004:** Quantification of MT#39998 viscotoxins by HPLC.

Viscotoxin	µg/mL
VTA1	75.629
VTA2	165.185
VTA3	221.442
VTA B	0
VT Tot	462.256

**Table 5 plants-14-02762-t005:** Peptides identified by untargeted proteomic experiments. *Viscum* proteins are shown in bold. The Appendix A show the number of peptides identified and their corresponding sequences (Appendix A).

Protein	Specie
ABC transporter G family member 30	Arabidopsis thaliana
Allene oxide synthase 2, chloroplastic	*Solanum lycopersicum*
Beta-galactoside-specific lectin 1	*Viscum album*
Beta-galactoside-specific lectin 2	*Viscum album*
Beta-galactoside-specific lectin 3	*Viscum album*
Calmodulin	*Bryonia dioica*; *Triticum aestivum*
Calmodulin-1	*Solanum tuberosum*
DEAD-box ATP-dependent RNA helicase 40	*Arabidopsis thaliana*
FBD-associated F-box protein At4g10400	*Arabidopsis thaliana*
F-box protein At3g19890	*Arabidopsis thaliana*
NAD(P)H-quinone oxidoreductase subunit 5, chloroplastic	*Morus indica*
Oxygen-evolving enhancer protein 1, chloroplastic	*Chlamydomonas reinhardtii*
Pentatricopeptide repeat-containing protein At1g18485	*Arabidopsis thaliana*
Peroxidase 23	*Arabidopsis thaliana*
Peroxidase A2	*Armoracia rusticana*
Putative heat shock protein 2 (Fragment) OS=	*Pseudotsuga menziesii*
Putative UPF0496 protein 5 O	*Oryza sativa* subsp. *indica*
Remorin 4.1 OS	*Arabidopsis thaliana*
Ribulose bisphosphate carboxylase small chain 1A, chloroplastic	*Arabidopsis thaliana*
Ribulose bisphosphate carboxylase small chain SSU1, chloroplastic	*Lemna gibba*
Spermidine hydroxycinnamoyltransferase 1	*Oryza sativa* subsp. *japonica*
Superoxide dismutase [Cu-Zn] OS=	*Ananas comosus*
Superoxide dismutase [Cu-Zn], chloroplastic	*Petunia hybrida*
Terpenoid synthase 9	*Arabidopsis thaliana*
Thionin OS	*Pyrularia pubera.*
Viscotoxin-1-PS	OS = *Viscum album*
Viscotoxin-A3 OS = *Viscum album;*	OS = *Viscum album*
Viscotoxin-B (Fragment) OS = *Viscum album*	OS = *Viscum album*

**Table 6 plants-14-02762-t006:** Viscotoxin quantification by HPLC-MS/MS in MRM. VT = viscotoxins; MT# = *V. album* mother tincture followed by batch number; pv = eluted viscotoxins; L-MT is a mother tincture produced only from viscum leaves; VC is Vischio Centofiori; IM is Iscador M; Co is Coragil; RA is Rubax Artro; VB is Vischio Biokyma; vd-MT = viscotoxin-deprived mother tincture. Data are reported as µg/100 µL.

VT	MT #39998	pv #39998	MT #40286	MT#40293	L-MT	VC	IM	Co	RA	VB	vd-MT #39998
**A**	2.4	5.4	2.31	1.69	0.63	1.23	6.3	0.83	0.75	0.23	-
**A3**	31.6	38.8	31.20	33.50	43.69	19.02	32.2	50.31	57.04	26.35	-
**C**	1.5	1.1	0.42	0.59	0.13	0.58	14.3	-	-	-	-
**A1**	11.5	10.5	28.60	21.52	17.09	12.30	10.2	0.56	0.52	0.21	-
**A1PS**	1.6	3.2	1.88	3.44	3.07	2.40	1.0	1.78	0.66	0.13	-
**A2**	24.2	21.1	19.50	27.30	17.60	21.60	26.2	35.20	27.60	10.52	-
**B**	4.2	3.8	4.70	4.10	3.70	3.60	9.5	1.69	1.00	0.25	-
ToT	**77**	**83.9**	**88.61**	**92.14**	**85.91**	**60.73**	**99.7**	**90.37**	**87.57**	**37.69**	-

**Table 7 plants-14-02762-t007:** IC_50_ values (% *v*/*v*) obtained for MT#39998 and vd-MT on HaCaT, BALB/c-3T3, SVT2 and HeLa cells after 48 h incubation.

Cell Line	MT#39998	vd-MT
HaCaT	2.6 ± 0.2	>5
BALB/c-3T3	1.6 ± 0.2	>5
SVT2	1.3 ± 0.8	5.0 ± 0.1
HeLa	2.1 ± 0.1	>5

**Table 8 plants-14-02762-t008:** Calculation of presumptive in vivo LD_50_ from in vitro IC_50_ data.

Cells	IC_50_ vol%	LD_50_ g/kg Bw
HaCaT	2.60	4.57
Hela	2.10	4.15
BALB/c-3T3	1.60	3.82
SVT2	1.30	3.53

## Data Availability

The original contributions presented in this study are included in the article/Appendix A. Further inquiries can be directed to the corresponding author.

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
