# Peer review of "Phytochemical Composition and Antioxidant Activity of a Viscum album Mother Tincture"

_plants, 2025, doi:10.3390/plants14172762_

Round 1

Reviewer 1 Report

Comments and Suggestions for Authors

This study conducted LC-MRM-MS and antioxidant experiments at the cellular level, but there are currently many issues that require revisions.

1.In the introduction, AreViscum album L., Viscum, and V. album arethe same plant?

2.In Table 1 (line 101), #40293 in Materials and Methodswas collected in March, but in the results and discussion (line 283)was in February.

3.#40293 prepared in March (line 293)has a more diverse range of phenolic compounds than #39998 prepared in November. Why was #39998 chosen for subsequent experiments instead of #40293?

4.Please check whether the cell viability experiment of the MTT method is in volume units or mass units? (line 381) .

5.The clarity of Figure 1 is insufficient; there should not be evaluative sentences in the title of Figure 1.

6.Figure 2 lacks a caption; Table 6 and Table 4lack thetitles.

7.Why is there no conclusion?

Author Response

Comment 1.In the introduction, AreViscum album L., Viscum, and V. album arethe same plant? Response The same species botanically speaking

Comment 2.In Table 1 (line 101), #40293 in Materials and Methods was collected in March, but in the results and discussion (line 283) was in February. Response The collection was made in different periods and the analyses were quite complex

Comment 3.#40293 prepared in March (line 293)has a more diverse range of phenolic compounds than #39998 prepared in November. Why was #39998 chosen for subsequent experiments instead of #40293? Response Phenolic content was not the only considered parameter.

Comment 4.Please check whether the cell viability experiment of the MTT method is in volume units or mass units? (line 381) . Response In volume units

Comment 5.The clarity of Figure 1 is insufficient; there should not be evaluative sentences in the title of Figure 1. Response Done

Comment 6.Figure 2 lacks a caption; Table 6 and Table 4lack thetitles. Comment done

Comment 7.Why is there no conclusion? Response Conclusion is now present

Reviewer 2 Report

Comments and Suggestions for Authors

Dear Authors,

Below are the comments that need to be completed in order for the manuscript to be accepted for publication.

Line 112: make the font uniform (keratinocytes)

Line 139: make the font uniform (USA)

Line 214: list the standards used for each group of analytes

Lines 235-238: make the font uniform

Line 244: to standardize markings throughout the manuscript (GC-MS. GC/MS)

Line 245: check HB-5 ms. The column with the specified label does not exist.

Lines 255 and 256: state what was used as an external standard

Loine 295: 3-O-glucoside; According to my knowledge, O should be written in italics. Correct throughout the manuscript.

Line 315: Please add a table caption.

Author Response

Comment 1 Line 112: make the font uniform (keratinocytes)   Response done

Comment 2 Line 139: make the font uniform (USA)   Response done.

Comment 3 Line 214: list the standards used for each group of analytes  Response done, see also in Supplementary, being the number of standards very high

Comment 4 Lines 235-238: make the font uniform   Response done

Comment 5 Line 244: to standardize markings throughout the manuscript (GC-MS. GC/MS)   Response done

Comment 6 Line 245: check HB-5 ms. The column with the specified label does not exist.  corrected

Comment 7 Lines 255 and 256: state what was used as an external standard Response done, see in particular the Supplementary, being the number of standards very high

Comment 8 Line 295: 3-O-glucoside; According to my knowledge, O should be written in italics. Correct throughout the manuscript.  Response Corrected

Comment 9 Line 315: Please add a table caption.  Response Added

Reviewer 3 Report

Comments and Suggestions for Authors

Dear editor, thanks for inviting me to review the manuscript entitled "Phytochemical Composition and Antioxidant Activity of a Viscum album Mother Tincture"

The manuscript sheds light on the phytochemical composition and antioxidant activity of a Viscum album mother tincture. The manuscript is well written, used advanced analytical techniques like LC-MRM-MS, HPLC etc for the extract analysis and its biological effects. The findings are highly significant for understanding the biological potential and safety of Viscum album extracts. However, some areas require clarification, and additional details would strengthen the manuscript.

Abstract is well written, have clear objective but could be revised slightly for consistency with the results. The introduction section should include a clearer statement of the research gap. In methodology section, the host tree (Malus domestica) and harvesting times are critical variables and author should add details for these factors and why this specific plant was selected and explain the conditions in detail. In line 127 the statement "Seguendo quanto riportato nell'Introduzione, questo saggio proporrei di metterlo dopo gli altri" should be explained or removed. Similarly, the sample size (n ≥ 3) should be mentioned in the figure captions for clarity. The author explained the differences in phytochemical composition due to harvesting time but did not discuss it in discussion section. It should be added in discussion section. On the same time in section 3.2.1 (HPLC Analysis): Table 4 does not have any caption that should be added for clarity. The author used appropriate formula for Estimation of LD50 but did not discuss its limitations of this approach. While the author wrote the discussion section in a much scientific way but there are few references and deeper discussion can improve this section. Similarly, author can add the clinical implications of the antioxidant activity especially against cited literature for cardiovascular and inflammatory conditions.

Specific comments are below

  1. Ensure all axes are labeled clearly and figures should have proper units for concentration.
  2. Tables should also have proper units like (mg/L) in column headers.
  3. Citations should be corrected for consistency.

The author can address above mentioned points to improve the manuscript.

Recommendations

Accept with Minor Revisions

Author Response

Comment 1. Abstract is well written, have clear objective but could be revised slightly for consistency with the results. Response Revised

Comment 2. It is necessary to remember that there is a limit of 200 words for the Abstract. Corrected

Comment 3. The introduction section should include a clearer statement of the research gap. Response Included.

Comment 4. In methodology section, the host tree (Malus domestica) and harvesting times are critical variables and author should add details for these factors and why this specific plant was selected and explain the conditions in detail. Response More information was added.

Comment 5. In line 127 the statement "Seguendo quanto riportato nell'Introduzione, questo saggio proporrei di metterlo dopo gli altri" should be explained or removed. Response Removed, sorry for the inconvenience

Comment 6. Similarly, the sample size (n ≥ 3) should be mentioned in the figure captions for clarity. Response Done

Comment 7. The author explained the differences in phytochemical composition due to harvesting time but did not discuss it in discussion section. It should be added in discussion section. Response Done

Comment 8. On the same time in section 3.2.1 (HPLC Analysis): Table 4 does not have any caption that should be added for clarity. Response Added.

Comment 9. The author used appropriate formula for Estimation of LD50 but did not discuss its limitations of this approach. Response Added adequate considerations.

Comment 10. While the author wrote the discussion section in a much scientific way but there are few references and deeper discussion can improve this section. Response References were added.

Comment 11. Similarly, author can add the clinical implications of the antioxidant activity especially against cited literature for cardiovascular and inflammatory conditions. Response This argument was already discussed in a previous paper, which is reported as reference

Specific comments are below

  1. Ensure all axes are labeled clearly and figures should have proper units for concentration.  Done
  2. Tables should also have proper units like (mg/L) in column headers.  Done
  3. Citations should be corrected for consistency.   Done

Round 2

Reviewer 1 Report

Comments and Suggestions for Authors

After the author's meticulous and careful revision, the requirements for publication have been basically met. There are still a few minor issues for the author's reference:

  1. Papers generally do not use the first person.
  2. Some details need to be carefully checked. As repeated 2.6 of 157in Line .
  3. Use fewer sentences in the titles of supplementary.

Author Response

1.In the introduction, AreViscum album L., Viscum, and V. album arethe same plant? The same species botanically speaking

2.In Table 1 (line 101), #40293 in Materials and Methods was collected in March, but in the results and discussion (line 283) was in February. The collection was made in different periods and the analyses were quite complex

3.#40293 prepared in March (line 293)has a more diverse range of phenolic compounds than #39998 prepared in November. Why was #39998 chosen for subsequent experiments instead of #40293? Phenolic content was not the only one parameter.

4.Please check whether the cell viability experiment of the MTT method is in volume units or mass units? (line 381) . In volume units

5.The clarity of Figure 1 is insufficient; there should not be evaluative sentences in the title of Figure 1.

6.Figure 2 lacks a caption; Table 6 and Table 4lack thetitles. done

7.Why is there no conclusion? Conclusion added